# Using genetic data to identify transmission risk factors: Statistical assessment and application to tuberculosis transmission

**Isaac H. Goldstein**[1], **Damon Bayer**[1], **Ivan Barilar**[2], **Balladiah Kizito**[3], **Ogopotse Matsiri**[3], **Chawangwa Modongo**[3], **Nicola M. Zetola**[4], **Stefan Niemann**[2]*, **Volodymyr M. Minin**[1]*, **Sanghyuk S. Shin**[5]*

**1** Department of Statistics, University of California, Irvine, California, United States of America, **2** German Center for Infection Research, Research Center Borstel, Borstel, Germany, **3** Victus Global Botswana Organisation, Gaborone, Botswana, **4** Augusta University, Augusta, Georgia, United States of America, **5** Sue & Bill Gross School of Nursing, University of California, Irvine, California, United States of America

* stniemann@yahoo.de (SN); vminin@uci.edu (VMM); ssshin2@hs.uci.edu (SSS)

**Data Availability Statement:** All code and data are available at: https://github.com/igoldsteinh/kopanyo_tp_code.

## Abstract

Identifying host factors that influence infectious disease transmission is an important step toward developing interventions to reduce disease incidence. Recent advances in methods for reconstructing infectious disease transmission events using pathogen genomic and epidemiological data open the door for investigation of host factors that affect onward transmission. While most transmission reconstruction methods are designed to work with densely sampled outbreaks, these methods are making their way into surveillance studies, where the fraction of sampled cases with sequenced pathogens could be relatively low. Surveillance studies that use transmission event reconstruction then use the reconstructed events as response variables (i.e., infection source status of each sampled case) and use host characteristics as predictors (e.g., presence of HIV infection) in regression models. We use simulations to study estimation of the effect of a host factor on probability of being an infection source via this multi-step inferential procedure. Using `TransPhylo`—a widely-used method for Bayesian estimation of infectious disease transmission events—and logistic regression, we find that low sensitivity of identifying infection sources leads to dilution of the signal, biasing logistic regression coefficients toward zero. We show that increasing the proportion of sampled cases improves sensitivity and some, but not all properties of the logistic regression inference. Application of these approaches to real world data from a population-based TB study in Botswana fails to detect an association between HIV infection and probability of being a TB infection source. We conclude that application of a pipeline, where one first uses `TransPhylo` and sparsely sampled surveillance data to infer transmission events and then estimates effects of host characteristics on probabilities of these events, should be accompanied by a realistic simulation study to better understand biases stemming from imprecise transmission event inference.

**Funding:** This work was supported by the National Institutes of Health (R01AI147336 to IG, BK, OM, CM, SN, VMN, SSS; R01AI097045 to NMZ). The funder had no role in study design, data collection and analysis, decision to publish, or preparation of the manuscript.

**Competing interests:** I have read the journal's policy and the authors of this manuscript have the following competing interests: SSS declares receiving consulting fees from Beckman Coulter and Hycor Biomedical on unrelated studies. All other authors have declared that no competing interests exist.

## Author summary

Factors that affect infectious disease transmission are poorly understood, which impede efforts to prevent the spread of infectious diseases. Recently, software packages have been developed to infer transmission histories of infectious disease outbreaks using data from infectious disease genetics and epidemiology. These software packages have been used as part of methods to identify individual characteristics that affect infectious disease transmission. We used computer simulation to explore whether a statistical pipeline using the software package `TransPhylo` can successfully identify individual risk factors for being an infection source in a realistic public health setting where only a small proportion of pathogens are sequenced. We simulated tuberculosis (TB) outbreaks with different odds of being an infection source for TB transmission between people living with and without HIV. We found that the `TransPhylo`-based pipeline consistently underestimated the odds ratio for the association between HIV and being an infection source for TB transmission. We then applied this method to data from a TB study from Botswana and found no evidence of an association between HIV and being an infection source for TB transmission. Identification of transmission risk factors may be difficult in settings with low sampling proportion for genetic data.

## Introduction

Better understanding of risk factors for transmitting infectious disease can help improve public health interventions. For example, HIV infection is associated with reduced bacterial load and lower probability of cavitary disease among people with tuberculosis (TB) [1]. This has led to the hypothesis that HIV infection may reduce the probability of onward transmission of *M. tuberculosis* among people with TB. [1, 2] Improved understanding of the effect of HIV on *M. tuberculosis* transmission could inform prioritization of scarce public health resources for contact tracing. Moreover, identifying host factors for transmission could provide valuable insights into the pathophysiology of the infectious disease.

In practice, it is rare to have perfect knowledge of who infected whom in an infectious disease outbreak, making study of risk factors for transmission difficult. Advances in methods for pathogen genomics have led to enhanced understanding of infectious disease transmission. [3, 4] Sequencing data can be compared among infected hosts to rule out direct transmission when substantial genomic differences exist. More recently, several methods have been developed for using genomic and epidemiological data to reconstruct transmission events and identify a putative infection source for each transmission event [5–12]. In general, these tools were originally designed for outbreak investigations, where capturing most, if not all, cases was feasible. Recently, researchers have used such methods on data collected in the context of passive disease surveillance as part of a statistical pipeline to determine the association between potential risk factors and being an infection source (transmitting an infection to at least one other host) [13–15]. Fig 1 illustrates this pipeline. Researchers first infer a timed phylogenetic tree from genetic data, use the phylogenetic tree to infer infection source status labels for study participants, and finally infer the odds ratio of interest using the inferred infection source labels as observed response variables.

While passive disease surveillance could capture a significant portion of infectious disease incidence, we expect a meaningfully smaller proportion of cases will be sampled as compared to an outbreak investigation study. It is worth pointing out that we expect the proportion of cases sampled to heavily depend on the resources of a public health system, as well as the

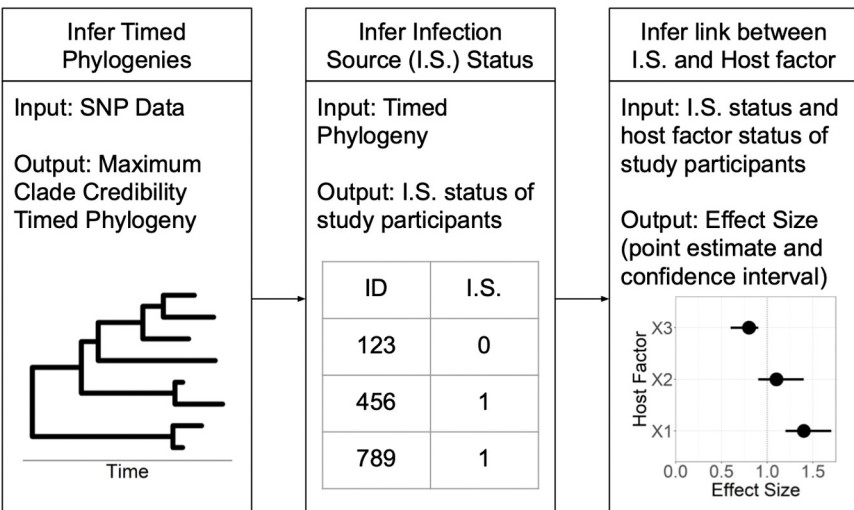

**Fig 1. Visualization of statistical pipeline evaluated in this paper.** Genomic data are used to construct timed phylogenetic trees, which are then used as inputs to infer transmission trees. Based on transmission trees, probability of being an infection source is derived, and then used as a response variable in an analysis of host factor associations.

prevalence of the disease being studied. In well resourced public health systems, the difference between passive surveillance and outbreak investigation may be small. Still, the performance of many of the transmission reconstruction methods in the context of passive surveillance is not at all clear. And while statistical pipelines relying on these transmission reconstruction methods have been used in practice, their ability to provide useful estimates of the associations between being an infection source and risk factors of interest remains largely untested. The goal of this study is to better understand the use of these pipelines in disease surveillance contexts by evaluating the performance of a set of statistical pipelines which use the commonly used `TransPhylo` software to reconstruct transmission events [5]. We evaluate bias and precision of the estimators of the odds ratios for probability of being an infection source. We also evaluate the power of statistical tests based on these estimators to reject a null hypothesis that the odds ratio is one, and coverage of corresponding 95% confidence and credible intervals. In particular, we evaluate the properties of using `TransPhylo` generated infection source labels and logistic regression, and `TransPhylo` infection source labels with two different statistical models that take into account measurement error. We accomplish this goal by simulating TB outbreaks with two classes of hosts (representing those living with and without HIV) with different probabilities of being an infection source where the underlying true odds ratios are known, and evaluating pipeline performance under a variety of true odds ratios and sampling designs. Finally, we apply our pipelines to real data to investigate the association between HIV infection and being an infection source for *M. tuberculosis* transmission.

## Materials and methods

### Ethics statement

The real data were collected for the Kopanyo Study, which received approval from the Health Research and Development Committee of the Botswana Ministry of Health and

Wellness, the U.S. Center for Disease Control and Prevention Institutional Review Board (IRB) (6291), and the University of Pennsylvania IRB. Informed consent was obtained from all participants.

## TransPhylo overview

TransPhylo is an R package which uses a statistical model and Markov chain Monte Carlo to produce a posterior distribution of transmission trees, graphs which describe who infected whom in an infectious disease outbreak [5]. TransPhylo uses timed phylogenetic trees of sampled cases in order to produce this posterior. One advantage of using TransPhylo for this task is that it is designed to allow for within-host evolution. It does this by "coloring" branches of the phylogenetic tree, where colors represent hosts, and allowing for color changes to occur anywhere on a branch, as opposed to just at bifurcations of the tree. Another advantage of TransPhylo is that it explicitly allows for outbreaks where not all individuals have been sampled. Using the posterior distribution of transmission trees, we can create a probability of being an infectious source for each sampled individual by calculating the proportion of posterior trees in which the individual transmitted the disease to another individual (regardless of whether the individual was sampled).

## Simulation set up

All simulated data sets were generated in R version 4.0.2 or 4.0.4 using the nosoi package version 1.0.3 [16, 17]. nosoi simulates an infectious disease outbreak by simulating individual cases which generate new cases in discrete units of time until either a specified time limit is reached or a specified total number of cases is reached. The behavior of cases is controlled through three parameters: the probability of the recipient ceasing to be infectious at the beginning of a time step, the number of susceptible persons an individual case encounters in a time step, and finally, the probability that, given a contact with a susceptible individual, an individual case will transmit the pathogen. For all simulations in this study, we assumed discrete time steps represented one month of real time, and chose model parameters to mimic a plausible *M. tuberculosis* outbreak (see A.1.1 in S1 Appendix for simulation parameters). In brief, $R_0$ was set to be 1.18, the mean latent period was 9 months among hosts that became infectious, and the mean infectious period was 3 months. All outbreaks used in this study lasted 8 years. In order to control the number of sampled recipients, we generated simulations at random, but only used outbreaks with a total number of between fifty and two thousand cases. This meant that many potential outbreaks were rejected as part of our sampling simulation process. While this rejection sampling has the potential to change the true odds ratio being estimated, we have found empirically this is not an issue for outbreaks lasting 8 years. In our simulation set-up, each outbreak represents a cluster of *M. tuberculosis* infected individuals to be analyzed jointly using TransPhylo, mimicking the common real world practice of clustering using SNPs by some threshold and generating separate timed phylogenies for each cluster. Henceforth we will refer to these outbreaks as clusters.

We simulated clusters with two types of individuals representing people living with and without HIV co-infected with *M. tuberculosis*. Based on data collected for a population-based epidemiology study of *M. tuberculosis* in Botswana (discussed below), we specified that on average 53 percent of cases would be people living with HIV. We had five primary simulation settings where the only changing parameter was the ratio of the probabilities of transmission given contact for two classes of cases. The settings were: probability of transmission given contact was 3.02 times higher among people living without HIV compared to that among people

with HIV, 1.75 times higher, equal, 1.75 times lower, or 3.33 times lower. All other parameters were identical for the two classes of cases. For the 3.02 setting, we will round down and refer to this as the 3 setting for the duration of the paper.

A single simulation consisted of fifty simulated clusters, where each cluster lasted eight years. From a single cluster, individuals were sampled from the last three years of the outbreak. Recipients were only sampled during their infectious periods, with the sampling time equally likely at any point in this time frame. For each cluster, we sampled 16% of eligible recipients for each of the last three years of the outbreak. Only simulations where all fifty clusters had at least two sampled individuals were accepted. In addition, because individuals could be active in multiple years, there were cases where there were not enough individuals in a particular year left to reach the percentage of 16%, simulations were likewise rejected in this scenario.

We chose this sampling scheme to reflect real world sampling conditions in our study setting, where data were collected as part of ongoing surveillance of an endemic disease over the course of multiple years, and where there was little chance of sampling recipients during their latent period. For example, due to under-diagnosis, the World Health Organization estimated that only 54% of people with new TB episodes in Botswana were reported in 2019 [18]. Using this proportion, we divided the number of *M. tuberculosis* sequences available by the number of estimated TB episodes (reported + unreported) to calculate the sampling proportion for our study. In more detail: we divided the number of diagnosed incidents of TB during the Kopanyo study (N = 4,331) by 0.538 to determine the total number of reported and unreported TB episodes (N = 8049). We then divided the number of sequences initially available for analysis (N = 1306) by 8049 to determine the sampling proportion of 16%. We continued to use this conservative sampling proportion estimate after increasing our sequenced samples to 1426. Thus, 16% reflects a reasonable sampling proportion one would expect when data come from passive surveillance of TB.

Once sampled, we generated timed phylogenetic trees for each cluster using `nosoi`. The trees from separate clusters represent the trees from separate transmission clusters which would be generated as part of a real data analysis. These timed phylogenetic trees served as the input for `TransPhylo` (version 1.4.4). For each simulation setting, we generated one hundred simulations.

In addition to our primary simulations, we conducted a number of secondary simulations to further investigate the operating characteristics of our statistical pipeline. For all secondary simulations, people living without HIV had a probability of transmission given contact that was set to be 1.75 times higher than people with HIV. We then varied the sampling proportion, increased total sample size, and decreased time from start of epidemic to start of sampling. To investigate the impact of increasing the sampling density, we increased the sampling proportion from 16% to 32% while keeping the sample size approximately the same by halving the number of clusters in a simulation (from fifty to twenty-five). To further investigate the consequences of increasing sampling density, we then increased the sampling proportion from 16% to 64%, again keeping the sample size approximately similar using only 13 clusters. To investigate the impact of an increase in sample size, we kept the sampling proportion at 16%, but doubled the number of clusters from fifty to one hundred. To investigate the impact of shortening the time from beginning of epidemic to beginning of sampling, we kept the sampling proportion at 16%, but increased the sampling window from the last three years to the last seven years of the simulation, while decreasing the number of clusters from fifty to thirty-five to approximately maintain the original sample size. All simulation settings are summarised in Table A1 in S1 Appendix.

## Using `TransPhylo` on simulated data sets

We used the multiple cluster functionality of `TransPhylo` to analyze all simulated trees from a single simulation simultaneously. The sampling proportion $\pi$ was allowed to vary among clusters, while all other parameters were shared across all trees. Based on empirical results from early simulations, for the primary simulations, the prior for the sampling proportion was set as a beta distribution with alpha equal to 1 and beta equal to 19. The other priors in `TransPhylo` cannot be changed by the user. `TransPhylo` assumes the number of individuals an infectious person will infect has a negative binomial distribution governed by two parameters $r$ and $p$. These parameters also describe the reproduction number, and so their priors control the prior for the reproduction number. $r$ has an exponential prior with rate parameter one, $p$ has a uniform prior over the interval zero to one. We found in practice that $p$ was not identifiable and fixed it at 0.5 for our analyses. Finally, the within-host effective population size also has an exponential prior with rate parameter one. The generation time and sampling time distribution were identical, with shape parameter 10, scale parameter 0.1. Parameters and their priors are summarised in Table A2 in S1 Appendix. We sampled 100,000 draws from the posterior distribution of transmission trees, discarding the first 50,000 as burn-in, and saving only every tenth sample. Probability of being an infection source was calculated as the proportion of posterior draws in which a particular case transmitted the disease. We considered a case to be an infection source if their inferred posterior probability of being an infection source was greater than 0.6. This same cutoff probability of 0.6 was used for the real data analysis as well. We have found the exact value of this cutoff makes little difference, as the distribution of probabilities tends to be extremely bi-modal, clustered around one and zero (see A.3 in S1 Appendix).

## Inferring odds ratios using infection source labels

All statistical inference was conducted in R (either version 4.0.2 or 4.0.4). We estimated odds ratios through logistic regression via the `glm` function in R. The mean of the response variable was the probability of being an infection source and the covariate was HIV status, where one indicates a person living without HIV, and zero indicates a person with HIV. That is:

$$\log\left(\frac{P(Y=1)}{1-P(Y=1)}\right) = \beta_0 + HIV^-\beta_1,$$

where $Y = 1$ when a case is labeled as an infection source and $Y = 0$ when they are not. For simulations, we used logistic regression using both the true infection source labels and the labels inferred by `TransPhylo`. True odds ratios for each simulation setting were calculated using Monte Carlo simulations (see A.2.1 in S1 Appendix). Our results suggested logistic regression using the `TransPhylo` inferred labels yielded sub-optimal inference due to mislabeled cases (i.e. `TransPhylo` inferred a case was not an infection source when in fact they were). This is a well-known phenomenon in statistics called measurement error, see Neuhaus (1999) for a clear introduction to the problem [19]. Many methods have been developed to adjust for this misclassification, we chose to include two as potential remedies to our statistical pipeline.

The first method is available in the `SAMBA` package by Beesley and Mukherjee (2020) (we used version 0.9.0) [20]. The `SAMBA` misclassification model assumes no misclassification of negative results (specificity is 1) and first infers the sensitivity of the response variable, then incorporates this sensitivity in the final logistic regression. To infer sensitivity, the user is required to input a fixed unconditional probability of the response variable being one, in our setting, this is the unconditioned probability that a case is an infection source. For simulations, the true unconditional probability from the sampled data was used. For the real data analysis,

the mean probability of being an infection source in the last three years of the cluster from one of the primary simulation settings was used.

The second method is a Bayesian approach we created to account for the case when specificity is not 1. For this model we assume the sensitivity and specificity for the response labels are known. In the model below $Y$ represents the observed response label whereas $Z$ represents the unobserved true response label.

$$\beta_0 \sim N(0, 1),$$
$$\beta_1 \sim N(0, 1),$$
$$\log \frac{P(Z_i = 1)}{1 - P(Z_i = 1)} = \beta_0 + HIV_i^- \beta_1,$$
$$Y_i \mid Z_i = 0 \sim \text{Bernoulli}(p = 1 - \text{Specificity}),$$
$$Y_i \mid Z_i = 1 \sim \text{Bernoulli}(p = \text{Sensitivity}).$$

For the simulations, Monte Carlo estimated sensitivity and specificity were used, rounded to two significant digits. For the real data analysis, the Monte Carlo estimated sensitivity and specificity from one of the primary simulations was used. There was little difference in estimated sensitivity and specificity between the different settings (**Figs A1 and A2 in** S1 Appendix). We used Hamiltonian Monte Carlo (HMC) to conduct Bayesian inference with this model using Stan through the `rstan` package (version 2.21.2). We generated 2000 posterior draws and discarded the first 1000 as burn-in. We used the median of the posterior distribution for point estimates and 95% Bayesian credible intervals as measures of uncertainty.

## Data collection

The real data used in this study was originally collected as part of a molecular epidemiology study of *M. tuberculosis* transmission in two regions of Botswana. Botswana has a high prevalence of HIV as well as *M. tuberculosis*. In 2017, 19.9% of people aged 15–49 were living with HIV, 300 of 100,000 people had TB, 48% of people with TB also had HIV [21]. All patients who were diagnosed with *M. tuberculosis* between 2013 and 2016 in the two regions were eligible for enrollment in the study. Of the 4331 enrolled participants, 2159 had culture-confirmed TB. Any participants who had unknown HIV status or who had had a negative result more than a year prior to sampling were offered HIV testing as part of the study protocol [22, 23].

## Sequencing

Of the *M. tuberculosis* DNA extracted from cultured isolates, 1671 (77%) were of sufficient quality to undergo sequencing using Nextera XT libraries. We performed WGS using Illumina Technology (MiSeq, NextSeq 500) generating 2x151bp paired end reads. All genomes were analysed with the MTBseq pipeline using the standard input values [24]. In short, reads were mapped to the *M. tuberculosis* H37Rv genome (GenBank ID: `NC_000962.3`) using BWA-mem and Samtools. Base call recalibration and realignment of reads around insertions and deletions was performed using GATKv3. Finally, variant calling was performed with Samtools mpileup and in-house scripts that employed thresholds for coverage and base quality. After sequencing, 1426 libraries were of high enough quality, defined by at least 50x coverage and 95% of the reference genome covered, to be used for further analysis. Average coverage of the good quality samples was 120x.

Concatenated sequence alignment of variants was produced by including those genome positions that fulfilled the aforementioned criteria for coverage and variant frequency in 95% of all samples. Repetitive regions (PPE/PE-PGRS genes), InDels, consecutive SNPs in 12bp

windows and genes implicated in antibiotic resistance were excluded. Phylogenetically informative SNPs were taken from existing literature. We excluded sequences with heterogeneous reads along a base position for 45—50 SNPs or greater, which is indicative of mixed infections, leaving 1347 sequences for analysis.

## Analysis of real data

Within each lineage, we calculated raw single nucleotide polymorphism (SNP) distances between all sequences and clustered sequences based on two cutoffs, either sequences with a SNP distance of five or fewer, or ten or fewer were clustered together. For our analysis, we excluded clusters with less than four sequences. We also excluded clusters with little genetic variation by excluding clusters where there were less than two times the number of sequences in the cluster minus two varying sites. For the five SNP analysis, we had 147 sequences from 21 clusters. For the ten SNP analysis, we had 388 sequences from 46 clusters to analyze.

We used BEAST 2 version 2.6.3 to create posterior distributions of timed phylogenetic trees with 10 million posterior draws for each of our clusters [25]. We then calculated maximum clade credibility trees to use as the inputs into `TransPhylo`, discarding the first ten percent of Markov chain Monte Carlo iterations. We used an HKY model with strict molecular clock and a constant effective population size model, and used an ascertainment bias correction for sequences of unknown length to account for the fact that analyzed sequences included only varying sites [26]. We used a uniform prior for the molecular clock with a lower bound $1 \times 10^{-8}$ and upper bound $5 \times 10^{-7}$ scaled by 4.2 million divided by the number of sites in the data for each cluster. The scaling was done to account for the fact that we used BEAST2 on SNPs, not the full genome. The upper and lower bounds were chosen based on work by Menardo et. al, and were chosen to be suitable for all TB lineages [27]. Priors on the frequency parameter, kappa and population size were all log normal distributions with parameter M set to 1.0 and parameters S set to 1.25.

For `TransPhylo`, shape and scale parameters for the generation time and sampling time distributions were fixed at 10 and 0.1 respectively. The multitree function was used to conduct inference simultaneously on all clusters generated by the same SNP threshold, with all parameters but the sampling proportion shared among all clusters. We used a beta distribution with alpha set to 1 and beta set to 19 as a prior for the sampling distribution. We drew 100,000 posterior samples, discarding the first 50,000 samples as burn-in and saving every tenth sample. We calculated the odds ratio for people living without HIV vs with HIV using the inferred `TransPhylo` labels using logistic regression, using the SAMBA package, and using the Bayesian model as described in a previous subsection.

## Reproducibility

All code and data needed to recreate the results of this study are available at https://github.com/igoldsteinh/kopanyo_tp_code.

## Results

### Simulated data

Table 1 shows summary statistics of the simulated clusters used in the primary simulation setting where people living with HIV were 1.75 times less likely to transmit infection given contact compared to people without HIV. Median cluster size was 116, with 95% of clusters ranging from 53 to 393 hosts. Latent periods for cases had a median of 9 months, while infectious periods lasted a median of 3 months. Median tree height was 5.9 years.

**Table 1. Simulation summary statistics for 8 year simulation.** Summary statistics of simulated infectious disease outbreaks from 100 simulations where probability of transmission given contact for hosts without HIV was 1.75 times larger than for hosts with HIV. *Infection Source—transmitting an infection to at least one other host.

| Metric | Median | [2.5% Q, 97.5% Q] |
|---|---|---|
| Cluster Size | 116 | [53, 393] |
| Samples per Cluster | 9 | [3, 34] |
| Latent Period (Months) | 9 | [4, 17] |
| Infectious Period (Months) | 3 | [1, 15] |
| $P(IS^*)$ | 0.69 | [0.64, 0.73] |
| $P(HIV^+)$ | 0.53 | [0.49, 0.56] |
| $P(IS\|HIV^-)$ | 0.77 | [0.71, 0.81] |
| $P(IS\|HIV^+)$ | 0.61 | [0.57, 0.67] |
| $P(HIV^+\|IS = N)$ | 0.65 | [0.58, 0.71] |
| $P(HIV^+\|IS = Y)$ | 0.48 | [0.43, 0.52] |
| Tree Height (Years) | 5.92 | [2.59, 7.42] |
| % TP Labeled IS | 19.47 | [17.11, 22.35] |

## Operating characteristics of statistical pipeline

For each simulation setting, we calculated the sensitivity and specificity of `TransPhylo` for identification of an individual as an infection source. Point estimates for each simulation setting are displayed in **Figs A1 and A2 in** S1 Appendix. In general, specificity was high, around 0.97, while sensitivity was low, approximately 0.28. Sensitivity increased when either the sampling density or the sampling window was increased.

Then, for each version of our pipeline and each simulation setting, we calculated frequentist coverage (proportion of simulations for which a 95% confidence/credible interval contained the true parameter–ideally it should be 0.95), proportion reject: the proportion of times that a null hypothesis that the odds ratio was one was rejected (with significance level 5%), percent bias (bias divided by the true odds ratio), and mean confidence interval width of 95% confidence intervals. Fig 2 shows the results from the primary set of simulations. The four pipelines tested for each simulation were: a reference pipeline where true infection source labels were known and used as inputs into a generalized linear model (Truth + GLM), a pipeline where `TransPhylo` labels were used as input into a generalized linear model (TP + GLM), a pipeline where `TransPhylo` labels were used as input into SAMBA (TP + ME1) for misclassification adjustment and a pipeline where `TransPhylo` labels were used as inputs in our Bayesian model for misclassification adjustment (TP + ME2). Across simulation settings, the Truth + GLM pipeline had coverage around 0.95, proportion reject above 0.95 (except in the case when the odds ratio was 1), percent bias near 0 and confidence interval width ranging from 0.17 to 2.7. For the TP + GLM pipeline, coverage varied depending on the simulation setting, ranging from 0.62 to 0.93. When the true odds ratio was not one, TP + GLM had proportion reject ranging from 0.54 to 1. Percent bias ranged from -16.76% to 48.34%, mean credible interval width was 0.33 to 2.28. The TP + ME1 pipeline had coverage and proportion reject similar to TP + GLM, with percent bias ranging from -55.83% to 195.66% and MCIW between 0.25 and 23.94. Finally, the TP + ME2 pipeline had coverage ranging from 0.79 to 0.98, proportion reject ranging from 0.44 to 1, percent bias from -30.52% to 118.51% and MCIW from 0.36 to 22.

The results of our secondary set of simulations (along with the original results from the 1.75 setting) are displayed in Fig 3. For the TP + GLM pipeline, increasing the percent of hosts sampled from 16% to 32% (Double Sampling Density) led to increased coverage and increased

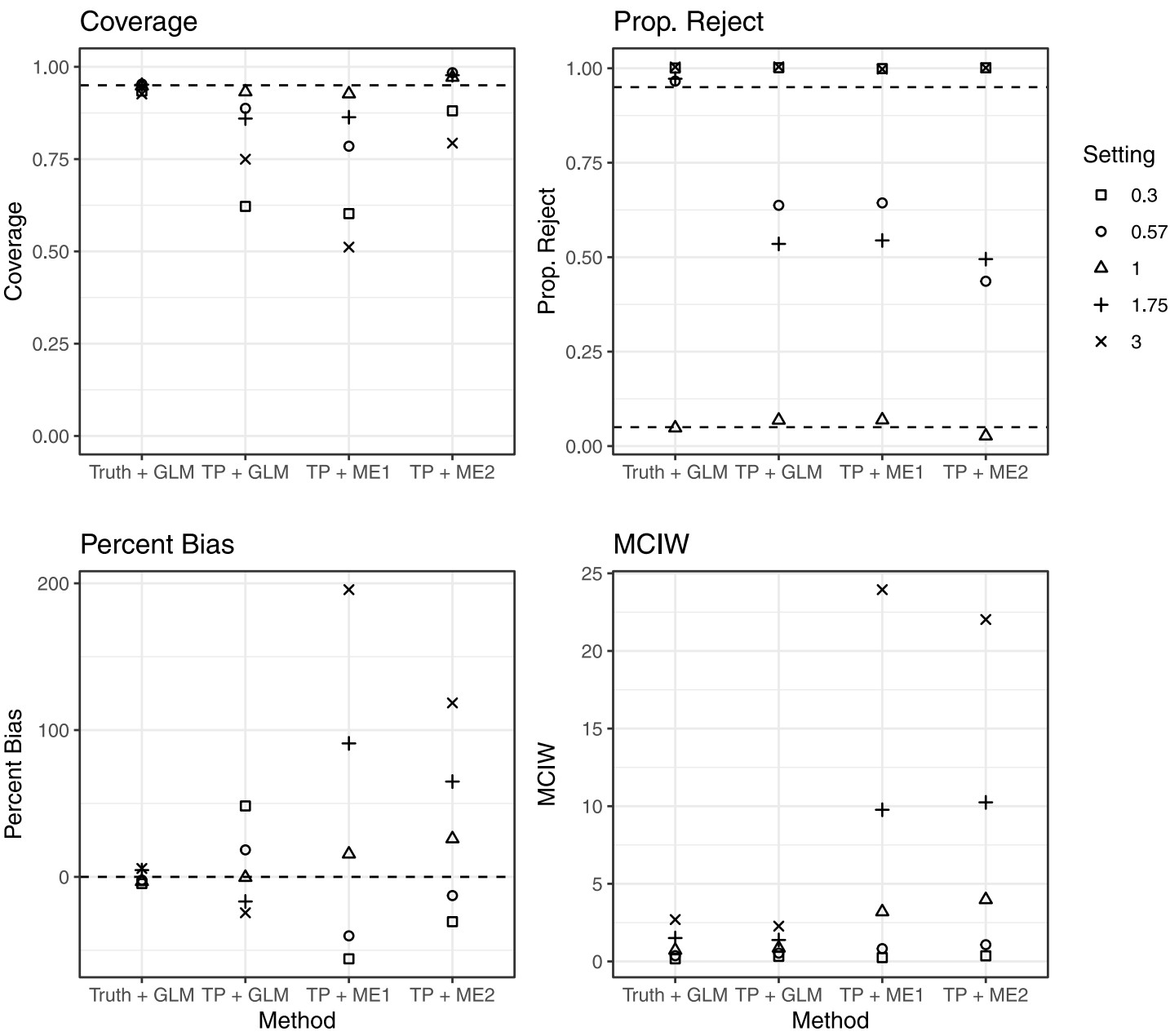

**Fig 2. Operating characteristics of statistical pipelines in primary simulation settings.** Truth + GLM is a reference pipeline where the true infection source labels are used as response variables in a generalized linear model. TP + GLM is a pipeline where infection source labels generated by `TransPhylo` are used as response variables in a generalized linear model. TP + ME1 is a pipeline where infection source labels are generated by `TransPhylo` and used as an input into a model from the `SAMBA` package, which allows for false positives. TP + ME2 again uses labels generated by `TransPhylo` as response variables into a model which allows for both false positives and false negatives. The settings denote different simulation settings, each value describes the true ratio of the probability of transmission given contact for cases living without HIV to cases with HIV. I.E., 0.57 means that the probability of transmission given contact for hosts without HIV was 1.75 times as small as the probability of transmission given contact for hosts with HIV. Coverage refers to proportion of simulations where 95% confidence intervals captured the true odds ratio. Prop. Reject refers to the proportion of simulations where a null hypothesis that the true odds ratio was one would be rejected, assuming significance level of 5%. Percent bias is bias divided by the true odds ratio, MCIW is mean confidence interval width.

proportion of scenarios where a null hypothesis that the odds ratio was one was rejected. Quadrupling the percent of hosts sampled to 64% (Quad Sampling Density) led to a decrease in coverage compared to both the default and the Double Sampling Density simulation, although the proportion reject remained comparable with Double Sampling Density

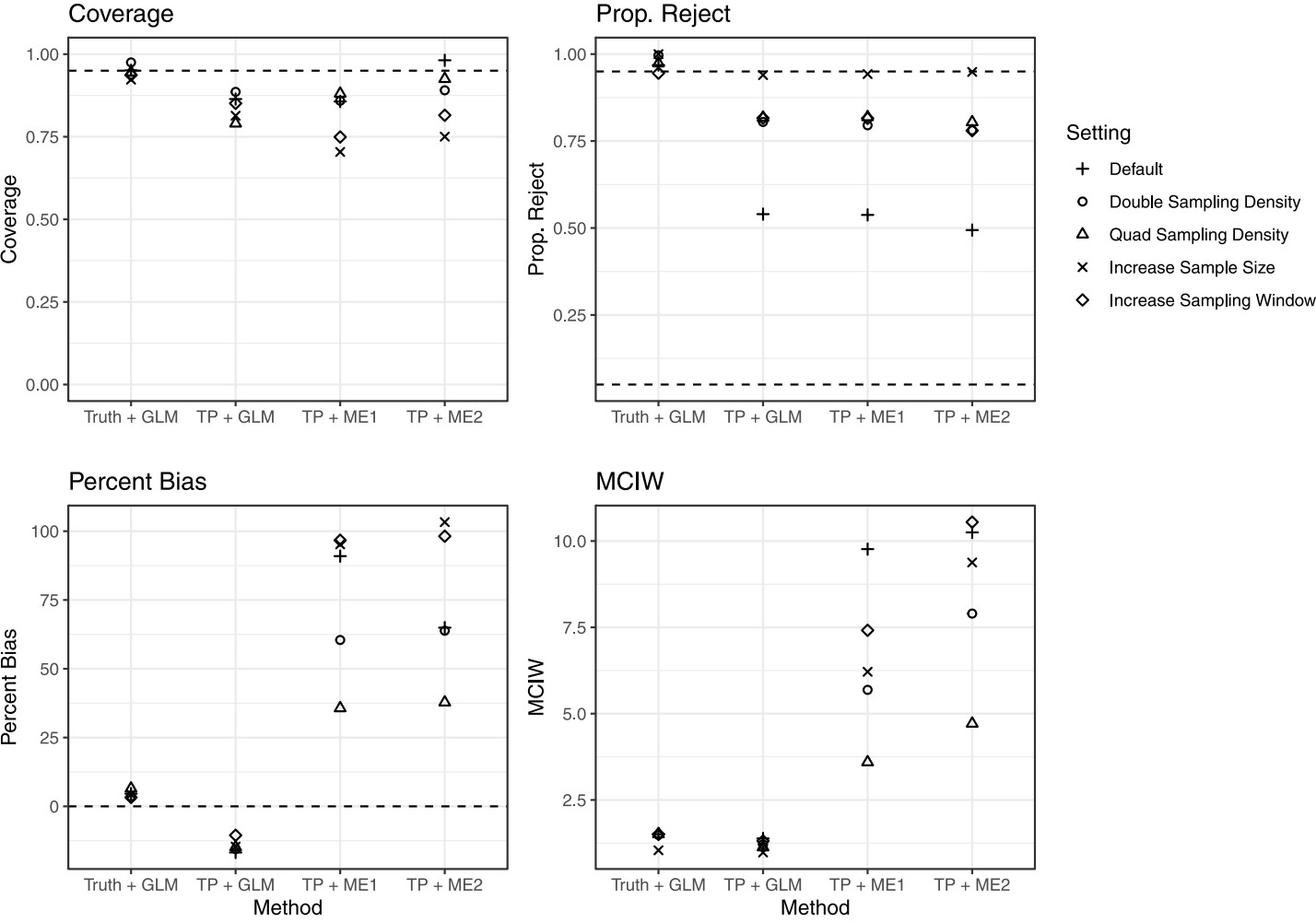

**Fig 3. Operating characteristics of statistical pipelines in secondary simulation settings.** See Fig 2 for x-axis and y-axis labels. The settings denote different simulation settings. For all simulations the probability of transmission given contact for cases living without HIV was 1.75 times as large as the probability of transmission given contact for cases living with HIV. Default refers to the simulation settings used in the primary simulation setting. In the Double Sampling Density setting percent of active cases sampled was doubled from 16% to 32%. In the Quad Sampling Density setting percent active cases sampled was quadrupled to 64%. In the Increase Sample Size setting the number of clusters sampled from was doubled from 50 to 100. In the Increase Sampling Window the sampling window was changed from the last three years of the simulation to the last seven years.

simulation, and the bias decreased. Increasing the sampling window to all but the first year of the outbreak slightly decreased coverage and increased proportion reject. Doubling the number of clusters analyzed from fifty to one hundred (Increase Sample Size) led to the lowest coverage of all settings but the highest proportion reject. For the two measurement error pipelines, the proportion reject improved in a similar manner as in the TP + GLM pipeline. However, the coverage changed in a different manner, with both the Increase Sample Sizes and the Increase Sampling Window having substantially lower coverage than the default setting.

## Real data results

Study participant characteristics from the Botswana study are summarized in Table 2. The vast majority of sampled TB bacteria belonged to lineage 4. More than half of study participants were male, and most participants were living with HIV.

**Table 2. Study participant characteristics.** Summary statistics of participant characteristics from the study of *M. tuberculosis* in Botswana.

|  | Count | Percent |
|---|---|---|
| **TB Lineage** |  |  |
| Lineage 1 | 83 | 6.4 |
| Lineage 2 | 69 | 5.3 |
| Lineage 3 | 12 | 0.9 |
| Lineage 4 | 1142 | 87.4 |
| **Sex** |  |  |
| Male | 719 | 55.1 |
| Female | 587 | 44.9 |
| **Age** |  |  |
| Under 18 | 80 | 6.1 |
| 18 to 35 | 693 | 53.1 |
| 35 to 55 | 466 | 35.7 |
| 55 to 75 | 67 | 5.1 |
| 75 and older | 10 | 0.8 |
| **HIV Status** |  |  |
| Without HIV | 590 | 45.2 |
| With HIV | 716 | 54.8 |

The four largest phylogenetic trees from each of our real data analyses (using five SNPs and ten SNPs as clustering cutoffs) produced by our statistical pipelines are displayed in Fig 4. From our genetic samples, we used BEAST2 to create a phylogenetic tree, then used `TransPhylo` to label tips of the tree as being infection sources or not infection sources and compare with their recorded HIV status. The bottom row have tips labeled by ID numbers, demonstrating the way trees change when inclusion criteria change. All participants from the five SNP tree (right) remain in the same ten SNP tree (left), with additional participants included because of the relaxed inclusion criteria.

Table 3 shows results from our real data analysis using clusters generated under a five SNP cutoff rule and a ten SNP cutoff rule. Mean cluster size and mean tree height increased marginally depending on which cutoff was used (increasing from 7 to 8.43 participants and from 6.86 years to 8.27 years), but both are similar to cluster size and tree height in simulated data (Table 1). Overall sample size was smaller than in simulation settings. In both analyses, between 11% to 12% of participants were identified as infection sources. Using any version of our pipeline, we fail to reject (at a 5% significance level) a null hypothesis that the odds ratio is one. Using either measurement error pipeline, the confidence intervals are larger than those produced by the TP + GLM pipeline. For the five SNP Analysis TP + GLM had confidence interval width of 2.39, while TP + ME1 had confidence interval width 17.36, and TP + ME2 had confidence interval width of 2.70. Using data from the 10 SNP analyses, all pipelines had shorter confidence intervals. The TP + GLM pipeline had a confidence interval width of 1.40, TP + ME1 had confidence interval width of 6.58, and TP + ME2 had confidence interval width of 2.15.

In addition, we used a generalized linear model with additional covariates for participant gender, smoking status, past history of TB infection and excessive alcohol use. The model had no covariates for which a null hypothesis the odds ratio was one could be rejected, and the point estimates and confidence intervals for the odds ratio for HIV status were similar to those reported in Table 3.

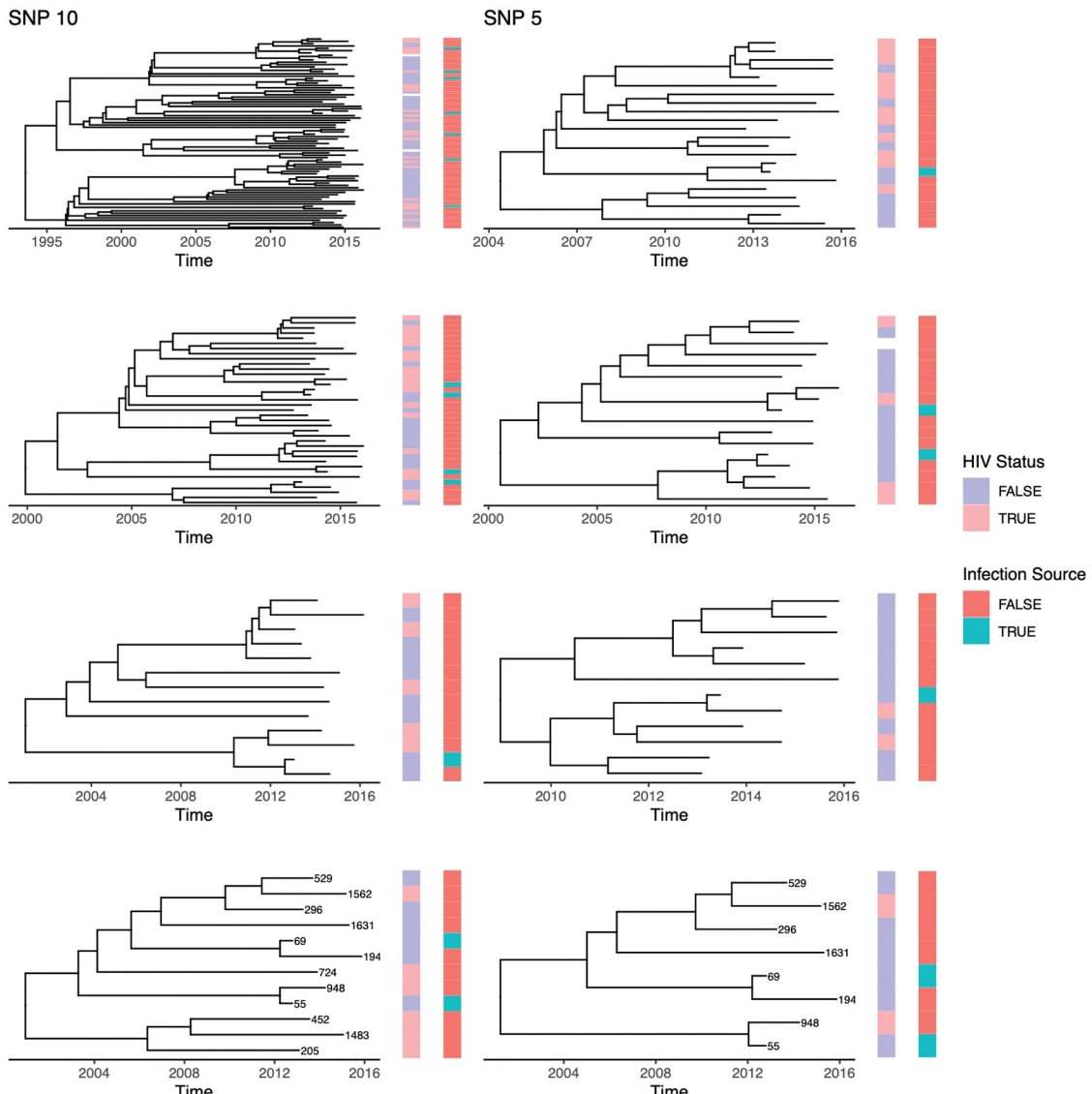

**Fig 4. Timed phylogeny with tips labeled by HIV status and inferred infection source status using `TransPhylo`.** Timed Phylogenies are generated in `BEAST 2`. Before timed phylogenies are generated, TB genomes are first clustered by SNP distance based on a cutoff of either five or ten SNPs. The trees generated under each cutoff are different, but related to each other, with larger trees overall using a cutoff of ten SNPs as opposed to five. The plot was made using R package `ggtree` [28].

## Discussion

In this study, we used simulations under realistic scenarios for passive TB surveillance to show that the statistical pipeline using `TransPhylo`-generated labels as response variables in a generalized linear model (TP + GLM) had poor frequentist properties for estimating the association between host characteristic and being an infection source. When the difference in probability of infection given contact was high between host categories, as in the 0.3 and 3 settings, TP + GLM results would lead one to correctly reject a null hypothesis the odds ratio was one more than 95% of the time, but due to bias and overly narrow confidence intervals, often failed to capture the true odds ratio. In the case of the 0.57 and 1.75 settings, TP + GLM results would lead to rejecting the null hypothesis less than 65% of the time. This poor performance is

**Table 3. Summary of analyses using five and ten SNP cutoffs.** Summary of analyses using five and ten SNP cutoffs. Mean tree height refers to the height of timed phylogenetic trees in years. TP + GLM refers to a statistical pipeline where infection source labels are first generated using `TransPhylo` and then used as response variables in a generalized linear model to calculate an odds ratio. The odds ratio is the odds ratio for the probability of being an infection source given the case is living without HIV as compared to cases living with HIV.

|  | 5 SNP Analysis | 10 SNP Analysis |
|---|---|---|
| Participants | 145 | 377 |
| Clusters | 21 | 46 |
| Mean Samples per Cluster | 7 | 8.43 |
| Mean Tree Height (Years) | 6.86 | 8.27 |
| Infection Sources | 16 | 43 |
| OR and 95% CI (TP + GLM) | 0.96 [0.33, 2.72] | 1.03 [0.55, 1.95] |
| OR and 95% CI (TP + ME1) | 0.88 [0.04, 17.4] | 1.10 [0.18, 6.76] |
| Mean OR and 95% CI (TP + ME2) | 0.80 [0.20, 2.90] | 0.96 [0.36, 2.51] |

partially explained by the sensitivity and specificity of `TransPhylo` for generating infection source labels. When used in our study setting, `TransPhylo` did not accurately label infection sources, leading to substantial under-counts of the true number of infection sources (**Figs A1 and A2 in** S1 Appendix). This measurement error resulted in poor estimation of the odds ratio of interest. The two measurement error models tested as part of a pipeline did not lead to improved coverage or higher rates of correctly rejecting the null hypothesis.

In all scenarios tested, the bias of using `TransPhylo` labels was such that the estimated odds ratio was always closer to one than the true odds ratio (Fig 2). When the true odds ratio was one, the pipeline performed well, with low bias, coverage near 0.95, and the probability of falsely rejecting null hypothesis near 0.05. This suggests that TP + GLM is a conservative pipeline, with bias towards failing to detect a significant effect, unless the effect size is large. We note that the simulations used in these studies created true timed phylogenies which were used as inputs into `TransPhylo`. In practice, these phylogenies must also be estimated from sequence data. Depending on the genetic diversity of available sequences, there may also be errors when creating the timed phylogeny. The errors in reconstruction of phylogenies could propagate further down the statistical pipeline, reducing its overall performance.

Our findings are consistent with a recent benchmarking study of computational methods for reconstructing TB transmission, which included `TransPhylo` and five other methods. [29] That study used simulations to show low sensitivity for correctly identifying transmission events under realistic low-TB burden scenarios for all methods tested. Another recent study used simulations to evaluate the use of genomic data and GLM for identifying risk factors for TB transmission, defined as genetic closeness (<2 SNPs). [30] That study found a consistent underestimation of the true odds ratio under multiple scenarios in low-TB burden settings. Our findings complement these prior studies by providing further insights into the limitations of using pathogen genomic data for understanding infectious disease transmission, with simulations and data based on a high TB burden setting.

The TP + GLM pipeline has now been implemented in multiple published studies using surveillance data of *M. tuberculosis* [13–15]. These studies assumed higher sampling proportions than those considered in our study. For example, Xu et. al. study of TB in Spain used a prior for the sampling proportion with plausible values ranging from 48% to 99% [13]. Depending on the surveillance setting, such high sampling density may indeed be plausible. In our study of TB in Botswana, based on the World Health Organizations estimates of the reporting rate for TB, we assumed a much lower sampling proportion [18]. We recommend carefully considering both the sampling scheme of any future studies using surveillance data

when conducting research using this pipeline, as well as the plausible effect sizes researchers wish to detect. Simulation studies with pre-specified effect sizes could be used as part of study design in order to help researchers decide what sampling protocol they should aim for in future studies. The code used in this study could be used as a starting point for conducting such simulation studies.

We found that changes to the sampling scheme could improve frequentist properties of the TP + GLM pipeline. For example, doubling the number of clusters under the 1.75 setting increased the proportion of null hypotheses rejected to 0.94 (Fig 3). However, just increasing the overall sample size did not improve coverage of confidence interval. While increased sample size decreased variance, and thus confidence interval width, it did not affect bias. In contrast, doubling the sampling density and increasing the sampling window both decreased bias and mean confidence interval width by a small amount. This led to improvements in proportion of null hypotheses rejected, and marginal improvements in coverage (increased sampling density) or only marginal losses in coverage (increased sampling window). The reduction in bias likely follows from improvements in the sensitivity of `TransPhylo` under these settings (**Figs A1 and A2 in** S1 Appendix). Less intuitively, we found that quadrupling the sampling density actually decreased the coverage due to narrower confidence intervals, even though the sensitivity of `Transphylo` improved. The bias decreased, as we would expect, but the mean confidence interval width decreased so much that coverage dropped. We speculate this is a consequence of the proportion of individuals labeled as infection sources. As this proportion nears 50%, we expect the estimated standard errors to narrow, because it is easier to estimate probabilities closer to 0.5. Our particular result is almost certainly dependent on the true effect size, that is, we would not expect that for all uses of this pipeline, a sampling proportion of 0.64 would decrease coverage as compared to a sampling proportion of 0.32. Our results suggest the behavior of this pipeline is unpredictable, a higher sampling density will not automatically result in improved coverage. When exact estimation of the effect size is important, we urge great caution when using this pipeline.

The improved performance under the increased sampling window suggests there may be room for improvement in the statistical model being used. Currently, `TransPhylo` assumes that the sampling proportion is constant throughout the period of the outbreak, while in our simulation scenarios sampling only took place in a fixed number of years of an outbreak (as it often would in a real life surveillance study). Future work could focus on changing this assumption in a `TransPhylo`-like model which would better match the the model likelihood to the data generation process. Though, again, we would urge caution, as improving the sensitivity of `TransPhylo` can lead to non-intuitive pipeline performance.

Applying the `TransPhylo`-based inference of who-infected-whom to a population-based TB study in Botswana, we failed to detect an association between HIV status and being an infection source for *M. tuberculosis* transmission. TB is the leading cause of death among people living with HIV [18]. The pathophysiology of HIV-TB co-infection is complicated as HIV-associated immunosuppression reduces *M. tuberculosis* bacterial load in HIV-infected hosts. This observation has led to the hypothesis that people living with HIV have a lower risk of onward TB transmission than people without HIV. [2] Epidemiologic studies investigating this hypothesis have produced conflicting results [15, 31, 32]. In light of our simulation study results, our failure to reject the null hypothesis may be due to inadequate statistical power for detecting an association of low to moderate effect sizes when sampling proportion is low. Additional studies using more densely sampled sequences are needed to provider stronger evidence regarding this putative association.

This study had several limitations which should be acknowledged. First, our simulations made several simplifying assumptions which make them imperfect representations of *M.*

*tuberculosis* outbreaks. In particular, we assumed latent and infectious periods were the same for hosts with and without HIV, and further assumed that sampling had no impact on infectiousness, whereas in the real world sampling would presumably be followed by treatment which would impact infectiousness. In our real data analysis, *M. tuberculosis* sequencing data were missing for majority of the enrolled participants. We assumed these data were missing completely at random. Moreover, the number of infection sources inferred by `TransPhylo` was small, which limited the ability to detect associations using GLM. Finally, our study assessed pipeline performance in the best case scenario: when timed phylogenetic trees are perfectly accurate. However, in the real world, phylogenetic trees must first be constructed from sequence data, a process which may affect pipeline performance. Conducting assessments of these pipelines using simulation engines which simulate sequence data is a useful area of future research.

Our findings provide new insights into the use of statistical pipelines that combine pathogen genomic and epidemiologic data to investigate host factors associated with onward transmission of infectious diseases. We recommend careful consideration of potential biases due to imprecise inference of transmission events when using these pipelines. Simulation studies based on the data collection context may improve the interpretation of the results of studies using these pipelines.

## Supporting information

**S1 Appendix. Additional text, tables, and figure for the manuscript.**
(PDF)

## Acknowledgments

We thank Xavier Didelot for helping us resolve issues using `TransPhylo` and Paul Bastide and Sebastian Lequime for their help using `nosoi`. We also thank Caroline Colijn, Jessica Stockdale and Vijay Jeevanantham Naidu for sharing their code using `TransPhylo` and `BEAST2`.

## Author Contributions

**Conceptualization:** Isaac H. Goldstein, Damon Bayer, Volodymyr M. Minin, Sanghyuk S. Shin.

**Data curation:** Ivan Barilar, Stefan Niemann.

**Formal analysis:** Isaac H. Goldstein.

**Funding acquisition:** Chawangwa Modongo, Nicola M. Zetola, Sanghyuk S. Shin.

**Project administration:** Balladiah Kizito, Ogopotse Matsiri, Chawangwa Modongo, Nicola M. Zetola.

**Supervision:** Chawangwa Modongo, Stefan Niemann, Volodymyr M. Minin, Sanghyuk S. Shin.

**Visualization:** Isaac H. Goldstein.

**Writing – original draft:** Isaac H. Goldstein.

**Writing – review & editing:** Damon Bayer, Ivan Barilar, Balladiah Kizito, Ogopotse Matsiri, Chawangwa Modongo, Nicola M. Zetola, Stefan Niemann, Volodymyr M. Minin, Sanghyuk S. Shin.

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
