## [Decision Letter · Decision Letter 0]

20 Jun 2022

Dear Dr. Shin,

Thank you very much for submitting your manuscript "Using genetic data to identify transmission risk factors: statistical assessment and application to tuberculosis transmission" for consideration at PLOS Computational Biology.

As with all papers reviewed by the journal, your manuscript was reviewed by members of the editorial board and by several independent reviewers. In light of the reviews (below this email), we would like to invite the resubmission of a significantly-revised version that takes into account the reviewers' comments.

We cannot make any decision about publication until we have seen the revised manuscript and your response to the reviewers' comments. Your revised manuscript is also likely to be sent to reviewers for further evaluation.

Sincerely,

Nicola Segata

Associate Editor

PLOS Computational Biology

Thomas Leitner

Deputy Editor

PLOS Computational Biology

Reviewer's Responses to Questions

**Comments to the Authors:**

Reviewer #1: This is a review for “Using genetic data to identify….” by Goldstein and collaborators. It is becoming common practice to go beyond the identification of transmission clusters in TB and reveal individual transmission links. This allows to understand risk factors differentially distributed between infectors and infectees. One popular approach is to use TransPhylo, a tool designed to transform phylogenetic trees into transmission trees where different probabilities for being an infector or an infectee can be assigned to a TB case. The authors review the use of TransPhylo and using both simulations and a real datasets explore the advantages and limitations of such an approach and how likely is that infectors/infectees are accurately predicted. Overall, the manuscript is clearly written and delivers and important message, as for any computational method in biology the users need to be aware of the limitations and the interpretability of the results. I believe it is a relevant message. I only have several question that I hope authors can clarify and help the reader understand better what is presented:

TransPhylo output assigns to each case a probability to be an infector/infecctee. This probability can be used to establish a threshold on how much you can rely in the prediction. This is important to avoid spurious assignment of infector/infectee status. However, it is not clear to me when and how this parameter has been used in the in the real dataset (there is a mention to it for the simulated dataset) and if so, which is the impact of changing the threshold in a real-world scenario?

The authors fail to find a negative association between HIV status and transmission. This is really interesting and very much in the line of other works using different approaches. But as the authors mention there are also reports for the contrary , and many times it is assumed PLHIV tend to transmit less. The failure to find an association maybe a real signal or just a limitation of the TransPhylo approach in the Botswana dataset. To rule out the second possibility I wonder if there is a problem on how an infectee is defined. For infectors you define those with >0.6 probability while for infectees you will expect to use only those with <0.4. Is that the case? Otherwise you will pool in the same infectees pot cases with a clear signal to be infectees with cases with no signal either in one direction or the other, this will dilute size of the studied effect.

Ascertainment bias and substitution rates. Please clarify if the analysis has been done only with SNP alignment or have been corrected for ascertainment bias (by the number of invariant sites). The second analysis is the correct one as it allows to use the known molecular clock rates calculated for Mtb. In addition, please clarify which molecular clock rate has been used and which is the rationale to use it (also because you have different lineages, do you think in using different rates for different lineages?)

Please, following the FAIR principles, deposit all the relevant data in repositories and provide accession (for sequence data) and DOI numbers for the metadata. Analysis must be reproducible and therefore please include the minimum information (sampling date, HIV status or any other variable needed to reproduce the analysis)

It is not clear to me which the final % of cases sequenced out of the total expected in the population? In general, given the extensive simulation work, can the authors suggests a coverage threshold from which TransPhylo can be applied with more confidence? This will help guide/design studies aiming to incorporate TransPhylo.

Reviewer #2: Attachment

Reviewer #3: Uploaded as an attachment.

**Have the authors made all data and (if applicable) computational code underlying the findings in their manuscript fully available?**

Reviewer #1: **No: **There is no mention to where the data has been deposited nor the associated metadata to run TransPhylo and reproduce the HIV analysis result (at the very least date and HIV status is needed)

Reviewer #2: Yes

Reviewer #3: **No: **Please include the code for Bayesian approach for inferring odds ratios

PLOS authors have the option to publish the peer review history of their article (what does this mean?). If published, this will include your full peer review and any attached files.

Reviewer #1: No

Reviewer #2: No

Reviewer #3: **Yes: **Benjamin Sobkowiak
---

## [Decision Letter · Decision Letter 1]

31 Oct 2022

Dear Dr. Shin,

We are pleased to inform you that your manuscript 'Using genetic data to identify transmission risk factors: statistical assessment and application to tuberculosis transmission' has been provisionally accepted for publication in PLOS Computational Biology.

Best regards,

Nicola Segata

Academic Editor

PLOS Computational Biology

Thomas Leitner

Section Editor

PLOS Computational Biology

Reviewer's Responses to Questions

**Comments to the Authors:**

Reviewer #2: The authors have satisfactorily answered each of the revision points.

Reviewer #3: The authors have addressed all of my comments adequately and, along with the changes made from other reviewer comments, I believe the manuscript to be much improved.

**Have the authors made all data and (if applicable) computational code underlying the findings in their manuscript fully available?**

Reviewer #2: Yes

Reviewer #3: Yes

PLOS authors have the option to publish the peer review history of their article (what does this mean?). If published, this will include your full peer review and any attached files.

Reviewer #2: No

Reviewer #3: No

---

## [Editor Report · Acceptance letter]

29 Nov 2022

PCOMPBIOL-D-22-00639R1 

Using genetic data to identify transmission risk factors: statistical assessment and application to tuberculosis transmission

Dear Dr Shin,

I am pleased to inform you that your manuscript has been formally accepted for publication in PLOS Computational Biology. Your manuscript is now with our production department and you will be notified of the publication date in due course.

With kind regards,

Anita Estes
